# Narrative Medicine: A Digital Diary in the Management of Bone and Soft Tissue Sarcoma Patients. Preliminary Results of a Multidisciplinary Pilot Study

**DOI:** 10.3390/jcm11020406

**Published:** 2022-01-14

**Authors:** Maria Cecilia Cercato, Sabrina Vari, Gabriella Maggi, Wioletta Faltyn, Concetta Elisa Onesti, Jacopo Baldi, Alessandra Scotto di Uccio, Irene Terrenato, Claudia Molinaro, Virginia Scarinci, Francesca Servoli, Cristina Cenci, Roberto Biagini, Virginia Ferraresi

**Affiliations:** 1Epidemiology and Tumor Registry Unit, IRCCS Regina Elena National Cancer Institute, 00144 Rome, Italy; cecilia.cercato@ifo.gov.it (M.C.C.); claudia.molinaro@ifo.gov.it (C.M.); 2Sarcomas and Rare Tumor Unit, IRCCS Regina Elena National Cancer Institute, 00144 Rome, Italy; elisa.onesti@ifo.gov.it (C.E.O.); virginia.ferraresi@ifo.gov.it (V.F.); 3Psychology Unit, IRCCS Regina Elena National Cancer Institute, 00144 Rome, Italy; gabriella.maggi@ifo.gov.it; 4Oncological Orthopaedics Unit, IRCCS Regina Elena National Cancer Institute, 00144 Rome, Italy; wioletta.faltyn@ifo.gov.it (W.F.); jacopo.baldi@ifo.gov.it (J.B.); allascotto@gmail.com (A.S.d.U.); roberto.biagini@ifo.gov.it (R.B.); 5Biostatistics and Bioinformatics Unit–Scientific Direction, IRCCS Regina Elena National Cancer Institute, 00144 Rome, Italy; irene.terrenato@ifo.gov.it; 6Digital Library «R. Maceratini», IRCCS Regina Elena National Cancer Institute, 00144 Rome, Italy; virginia.scarinci@ifo.gov.it (V.S.); francesca.servoli@ifo.gov.it (F.S.); 7Digital Narrative Medicine (DNM s.r.l.), 00161 Rome, Italy; cristina.cenci@dnmteam.com

**Keywords:** narrative-based medicine, digital diary, digital narrative medicine, personalized medicine, personalized care, bone and soft tissue sarcoma, multidisciplinary team care

## Abstract

Background. Guidelines for the implementation of narrative medicine in clinical practice exist; however, in Italy, no standard methodology is currently available for the management of oncological patients. Since 2017, at the “Regina Elena” National Cancer Institute, studies using “digital narrative diaries” (DNMLAB platform) have been carried out; this article focuses on a pilot, uncontrolled, real-life study aiming to evaluate the utility of DNM integrated with the care pathway of patients with bone and limb soft tissue sarcomas. Methods. Adult patients completed the diary during treatment or follow-up by writing their narrative guided by a set of narrative prompts. The endpoints were: (a) patients’ opinions about therapeutic alliance, awareness, and coping ability; (b) healthcare professionals’ (HCPs’) opinions about communication, therapeutic alliance, and information collection. Open- and closed-ended questions (Likert score: 1–5) were used to assess the items. Results. At the interim analysis of data from seven patients and five HCPs, DNM was shown to improve: (a) the expression of patients’ point of view, the perception of effective taking charge, disease awareness, and self-empowerment (score: 4.8/5); (b) patients’ communication, relationships, and illness knowledge (score: 4.6–4.8/5). Conclusions. The preliminary results supported the need to integrate patients’ narratives with clinical data and encourage further research.

## 1. Introduction

Narrative medicine was first described about 30 years ago as a new approach to the patient/clinician relationship [1]. Kleinman previously introduced a distinction between disease, which refers to the medical definition of a pathology, and illness, which refers to a patient’s subjective experience. Then, the role of narration was systemized as narrative medicine or narrative-based medicine by Rita Charon, who described the philosophical and theoretical context at the base of its application in daily clinical practice [1,2]. The use of narration was therefore introduced into medical practice, overcoming the traditional application usually limited to the psychological and psychiatric sciences.

It is well known that narration may have a therapeutic potential in itself; moreover, in collecting and interpreting information about a patient’s experience of an illness, it can provide healthcare professionals with data useful for clinical purposes, from the formulation of a more accurate and timely diagnosis to the personalization of care [3,4,5,6].

In 2014, the Italian institution, “Istituto Superiore di Sanità”, published recommendations for the implementation of narrative medicine in the management of rare and chronic degenerative diseases [7]; even though several studies introducing storytelling in the healthcare environment have been carried out since then, in Italy, no standard methodology is currently available for the management of oncological patients.

Nowadays, the digital setting is suitable and comfortable for use in patients’ self-narration. The digital era has led to the extension of oneself into a virtual space, resulting in changes to communication patterns. Digital storytelling is emerging as a new way to shape a narrative. In fact, more people than ever share their stories on social media by uploading pictures, tagging links with comments, writing little blurbs, and finding the means to express themselves. Done properly, storytelling can be a powerful, evocative, and emotional way of communicating themes and stories, often touching us in deeper ways than usual. 

An innovative and nonprofit start-up (DNM s.r.l.) created a digital platform (DNMLAB), designed to obtain guided narratives from patients during their path of care [8]. Since 2017, at the “Regina Elena” National Cancer Institute in Rome, several pilot studies using “digital narrative diaries” have been carried out with the aim to improve the oncological clinical practice for cancer patients in different settings of care [9,10,11]. The results provided data supporting the need to integrate patients’ narratives with clinical data and encourage further research [10,11].

Sarcomas are rare tumors; as a matter of fact, the latest AIRTUM (“Associazione Italiana Registri Tumori”) Working Group report on rare cancers indicated that in 2015, there was an estimated incidence of about 5900 new cases of soft tissue sarcomas and about 500 new cases of bone sarcomas in Italy [12]. The overall age incidence pattern for bone sarcomas is bimodal, with peaks at ages 10–19 and 65+. Of the three most frequent subtypes, osteosarcoma and Ewing’s sarcoma of the bone have their highest incidence at ages 10–19, while for chondrogenic sarcomas, the incidence is greatest at ages 65+.

In particular, patients affected by bone sarcomas represent a population with a high care need. The presence of several critical issues such as: young age at diagnosis, the aggressiveness and long duration of treatments, the management of mutilating surgery outcomes, as well as often inauspicious prognoses, require truly ‘global’ care. A multidisciplinary team approach, capable of creating solid therapeutic alliances, is required; the care team has to deal with situations of high emotional impact and has to support the patient’s process of self-awareness and appropriation of the disease condition. This context represents a setting in which the application of narrative medicine tools can bring significant benefits to the treatment process.

### 1.1. Digital Narrative Medicine: A Comprehensive Cancer Center Experience 

In 2017, we started a preliminary study at the “Regina Elena” National Cancer Institute with the aim to evaluate the feasibility, practicability, and self-assessment utility of the digital narrative medicine methodology from both patients’ and healthcare professionals’ (HCPs’) perspectives [10,11]. The DNMLAB gave a patient access to a protected personal area, called a “diary”, where he/she could write his/her narrative, guided by a set of narrative prompts made by the team.

Patients with breast or colorectal cancer receiving chemotherapy in the Oncology Department and patients with a solid tumor receiving radiotherapy in the Radiotherapy Department were included. Eight HCPs (two oncologists and six nurses) trained in the digital application of narrative medicine were allowed to participate. The compliance to the study was high: 31/46 invited patients (67%) participated. Gender, age, education, type of cancer, and treatment did not affect patients’ participation; the customary use of informatics and a predisposition to talk about oneself were the main drivers toward participation. The majority of participants were female (84%), with an average age of 52.5 years old (range 31–79). The results pointed out that patients related the utility of the platform to the increased ability to tailor treatments and to the possibility to express one’s own point of view; improvements in disease awareness and self-empowerment were also described. The strongest advantages reported by the HCPs were: the opportunity to disclose relevant individual data for treatment, otherwise not detectable, as well as the strengthening of communication and of the care relationship. 

### 1.2. Digital Narrative Diary in the Management of Sarcoma Patients

In 2020, on the basis of our previous experience, the pilot study entitled “Application of narrative medicine in the treatment of sarcoma patients” (AMENAS) began. The objective of this study, which is still ongoing, is to evaluate the utility of the narrative digital diary integrated in the care pathway of patients with bone and limb soft tissue sarcomas from both the patients’ and the healthcare professionals’ perspectives. This article reports the preliminary results and considerations from this pilot study.

## 2. Materials and Methods

### 2.1. Study Design 

This is a preliminary, open, uncontrolled study in the real-life setting. The study was approved by the Ethical Committee of IRCCS Regina Elena, Rome, Italy. Patients gave written informed consent to participate in the study, to use digital narrative medicine (DNMLAB), and to the use of data for research and assistance.

### 2.2. Patients

Patients who were consecutively referred to the Outpatient Sarcoma and Rare Tumor Unit with the diagnosis of sarcoma, at any stage of disease, who were candidates for chemotherapy treatment (+/− loco-regional treatment) or follow-up, were included. Eligibility criteria were the following: age ≥18 years, knowledge of the Italian language, availability of an electronic device, and an e-mail address. Exclusion criteria were: poor compliance, the presence of psychiatric disorders, and the presence of severe cognitive deficits. Patients’ general characteristics are reported in Table 1. 

### 2.3. Healthcare Professionals 

Six HCPs (three oncologists, a surgeon, a psychologist, and a case-manager nurse) trained in the application of narrative medicine participated. The entire team also included other professionals with expertise in the application of narrative medicine in research projects and qualitative research: an oncologist–epidemiologist, an anthropologist, a literature graduate, and librarians.

### 2.4. Intervention 

The HCPs invited the patients, providing them with an e-mail access to the diary, monitoring them by reading their stories, and then sharing and using these during the scheduled visits to personalize their treatment. Patient access to the digital diary was granted only upon invitation by an HCP, and the system met all the criteria of healthcare data protection required by the Privacy Authority and European regulation. The patients entered the platform via a computer or smartphone. The DNMLAB gave the patient access to a protected personal area, called a “diary” (DNM), where he/she could write his/her narrative, guided by a set of narrative prompts made up by the team. The prompts, conceived according to the patient’s path of treatment, were presented in sequence as the patient told his/her narrative and had a character limit (2000 to 3000). The patient could decide to ignore a proposed prompt and to integrate the narration with free observations. Each healthcare professional examined the text and discussed it among the team and with the patient during the scheduled visits. 

The study duration is fifteen months. The collection of narratives will take place over 12 months. 

The endpoints of the study were the feasibility (easy to access, time to use, and satisfaction) and utility (communication and relationship, therapeutic alliance, illness/disease knowledge, awareness, self-confidence, and empowerment) of the use of DNMLAB. The items were assessed by both HCPs and patients through a questionnaire with open and closed questions (Table 2 and Table 3). Open questions included comments and suggestions. Answers to closed questions were scored on a 5-point Likert scale, from 1 = complete disagreement to 5 = complete agreement [13]. The questionnaire was administered to the patients at the end of the collection of narration; HCP assessment will be required twice, at 6 months and at the end of the study.

A mixed qualitative–quantitative analysis methodology was used, including basic content methods (i.e., theme category and word cloud).

## 3. Preliminary Results

Preliminary analysis at 9 months indicates that 7 out of 20 enrolled patients concluded the narrative path and completed the final feasibility and utility assessment questionnaire. They are mainly males (M/F = 1.3) with a median age of 45 years old (range 31–54). The cancer site distribution is: bone sarcomas (57%) and limb soft tissue sarcomas (43%).

The seven patients’ final assessment questionnaire for the feasibility and utility of the narrative medicine diary was analyzed (Table 2). The average scores were generally high (>4/5) for the considered items. The highest values (≥4.7/5) regarded the possibility to express one’s own point of view, the perception of effectively taking charge, and the improvement of disease awareness and self-empowerment. Patients’ free comments included: “This tool is helpful for the patient to create a care relationship with all the professional team” (male, 52 years old); “the teamwork among all professionals greatly helps reassure the patient” (female, 38 years old); “a regular and constant use of this tool is supportive and useful for the patient” (female, 35 years old).

Out of six HCP participants, five of them completed the intermediate assessment. The surgeon did not carry out the assessment because he was poorly involved in the management of these patients during the study period. The feasibility items’ scores were generally good and included the improvement of clinical examination quality; however, lower scores regarded “time management” and “length”. These data indicate that the use of the diary did not significantly affect clinicians’ time management and the length of the scheduled visits.

According to HCPs’ evaluation of the utility items, the tool offers the opportunity to disclose relevant individual data that are otherwise not detectable (4.6/5); furthermore, it strengthens communication and relationship/alliance (4.8/5), not only between a patient and HCP but also among the members of the care team (Table 3). “The use of this digital diary allowed me to learn about some aspects of patients’ daily life that I was not aware of. This has led me to be more emphatic towards them; however, the emotional impact of this diary on healthcare professionals deserves to be carefully assessed” (oncologist, female, 36 years old).

## 4. Discussion

The preliminary data of this pilot study do not allow in-depth analysis; however, in light of our experience and considering data from the literature, several aspects deserve to be highlighted and deepened. For a better focus on the different topics, the discussion is organized into separate sections. 

### 4.1. Training and Education 

A narrative is a source of knowledge and requires learning techniques for its use: “Nonnarrative knowledge attempts to illuminate the universal by transcending the particular; narrative knowledge … attempts to illuminate the universals of the human condition by revealing the particular” [3]. In our experience, for a systematic narrative approach in clinical practice, not only training to improve narrative skills among physicians but also a change in their perspective towards care is truly required. The “Regina Elena” National Cancer Institute, according to the Organization of European Cancer Institutes (OECI) standardizing criteria for a comprehensive cancer institute, adheres to a patient-centered care model, an approach focused on individuals, aiming to practice personalized medicine for a better diagnosis and treatment. A program based on training and research projects, aiming to introduce a narrative model of clinical approach, was first introduced in 2015 when a story-sharing intervention started [9]. The intervention, named “Raccontami di te” (“Tell me about you”), addressed patients, their relatives and HCPs and was based on the sharing of individual stories with the aim to improve knowledge about the role of self-narration as well as the listening attitude. In order to increase the efficacy of care by improving narrative competence and careful listening, the project developed a communication strategy based on the promotion of reflexive writing among HCPs, patients, and caregivers. Training courses for HCPs, text analyses, story-sharing meetings, and the publication and distribution of the stories were used, so that a fertile ground for further studies applying narrative-based medicine was created. 

The preliminary results of this study pointed out, from the HCPs’ perceptives, the usefulness of this tool applied to clinical practice, particularly when considering the improvement of the quality of care, by deepening patients’ knowledge and strengthening the therapeutic alliance; on the other hand, their awareness of the need to gain specific skills to manage the “emotional impact” that the narratives of illness from young patients (often peers!) can generate emerged.

From the authors’ point of view, the time is ripe for the introduction of narrative medicine curriculum in the degree courses of HCPs, in order to gradually create a new framework within current medical practice [14,15,16].

### 4.2. Complexity and Multidisciplinarity

“The science of complexity has suggested an alternative model in which the disease, as well as the patient’s general well-being, is the results of a complex interaction between various elements of the entire system, dynamic and unique, of the individual” [17]. Complexity is the framework in which the alliance between physicians and their patients is achieved, and the ability to communicate is the basic assumption [17]. Narrative-based medicine aims to overcome the reductionist way of seeing the patient conceived in biomedicine, by perceiving patients as a whole, complex person.

This holistic clinical approach implies the concept of multidisciplinarity and transdisciplinarity of care, so natural, social, and health sciences are integrated [18,19]. Sarcomas are included among rare tumors, characterized by delayed diagnosis and often inauspicious prognoses due to their high metastatic potential (especially osteosarcoma and Ewing’s sarcoma) [12]. Bone sarcomas represent severe and disabling conditions with a low prevalence in the population; therefore, patients, who are often very young, and their families may have a perception of intense isolation and a sense of powerlessness. The treatment of these tumors usually foresees intensive and long-lasting chemotherapy treatments before and after aggressive surgical interventions, plus or minus radiotherapy [20]. The long time spent in hospitals to periodically carry out medical therapies (about 10–12 months, even for localized disease) can generate a deep sense of detachment from “normal” life and a difficult return to relational, social, and work settings. Moreover, the surgical approach can sometimes lead to disabling outcomes such as limb amputation and permanent disabilities that can inevitably change the planning of private and work life, especially for younger patients. To take care of these patients, a multidisciplinary team approach, capable of creating solid therapeutic alliances, is required; the care team has to deal with situations of high emotional impact and has to support a patient’s process of self-awareness and appropriation of the disease condition. In particular, in young people, the disease breaks into a phase of the life cycle in which the construction of one’s personal identity is not yet complete. The impact of the disease and the outcome of medical treatments on significant areas such as fertility, sexuality, body image, quality of life, and role in the family and society can make this evolutionary passage difficult and threaten the construction of adult planning. It is because we are aware of the complexity of the pathology and of the psychosocial problems involved in its treatment, that we think that tools such as narrative medicine can constitute an added value. 

The approach to the complexity requires a plurality of viewpoints. DNMLAB was created by a team of anthropologists and psychologists with input from doctors and experts in narrative medicine. It was used in the management of many clinical conditions, such as cardiovascular diseases and epilepsy, and other authors reported significant benefits in the treatment process [21,22,23]. In each study, the whole team conceived the narrative prompts according to the context, the illness/disease condition, the path of care and contributed to the text analysis. 

The preliminary results of this pilot study suggest that the use of DNMLAB improves communication and relationship, not only between patients and HCPs, but also among the members of care team themselves. The utility of the digital narrative medicine methodology tool was positively assessed in the patients’ perspective, who recognized its value in “creating a care relationship with all the professional team”, thus implementing a concrete patient-centered care model.

### 4.3. The Role of the Governance 

The application of qualitative techniques and narrative approach offers several advantages, including an improvement of the relationship between the citizen and the healthcare system, avoiding the waste of resources for conflict and optimizing assets [24,25].

The diary is appreciated as a tool for the application of narrative-based medicine and is suitable for clinical practice processes [10,11,21,22,23]. We conducted a further qualitative study among healthcare professionals, aimed at exploring the impact of this methodology on the perception of one’s own role [26]. This study pointed out that the narrative medicine approach engages HCPs’ personal experience and emotional resonances, leading them to redefine the values of health and disease. Moreover, the narrative approach must be considered in light of the relational systemic theory, taking the organizational development of the institutions into account. 

From the preliminary results of this study, a specific request for support by the institutions emerged: the need to “carefully” assess and manage the HCPs’ “emotional impact” as well as to consider patients’ narration as an integrated part of the care time management.

The systematic application of a narrative-based medicine therefore requires, on the one hand, the acquisition of skills by HCPs, and on the other hand, the adaptation of the model in which care is provided, including the organizational structures but also the values and the professional culture of work [26,27].

## 5. Conclusions and Future Perspectives

From the previous experiences, the digital narrative diary is appreciated as a tool for the application of narrative-based medicine, comfortable for use in self-narration by the patients and suitable for clinical practice processes in oncology as well as in other settings of care. This article focused on a pilot study we conducted to evaluate the utility of the diary integrated in the care pathway of patients with bone and limb soft tissues sarcomas. The preliminary results suggested that the use of DNMLAB offers patients the possibility to express their own points of view, increases the perception of effectively taking charge, and improves disease awareness and self-empowerment. Furthermore, according to the preliminary HCP assessment, it strengthens communication and relationships, not only with patients, but also among the members of the multidisciplinary care team themselves.

A narrative-based clinical approach could increase the quality and appropriateness of care. Further efforts and research are required to define the appropriate programs to apply this approach to the health sector.

## Figures and Tables

**Table 1 jcm-11-00406-t001:** Patients’ general characteristics.

Patient Characteristics	Participants *n* (%)
Evaluable/participating patients	7/20 (35)
Gender (male/female)	4/3 (57/43)
Median age, years (*range*):	45 (*37–54*)
Cancer site: histotype	
Bone: Ewing’s sarcoma (1), osteosarcoma (1), chondrosarcoma (1)	3 (43)
Limb soft tissue: myxoid liposarcoma (2), pleomorphic sarcoma (1), undifferentiated sarcoma (1)	4 (57)
Current treatment:	
Follow-up: NED after combined treatment	3 (43)
CT pre-surgery	2 (28.5)
CT for advanced/recurred disease	2 (28.5)

**Table 2 jcm-11-00406-t002:** Evaluation of the digital narrative diary by patients.

Patients’ Final Assessment	Likert Score (*N* = 7) Mean–SD	
**Feasibility**		
Diary friendliness	4.57	0.53
Diary immediacy and comprehensibility	4.57	0.53
Adequacy of one’s own computer skills	4.86	0.38
Opportunities to express oneself	4.86	0.38
Opportunities to provide personal information otherwise difficult to communicate	4.29	0.76
**Utility**		
Possibility to express one’s own point of view	4.71	0.49
Perception of effective taking charge	4.86	0.38
Improved awareness	4.86	0.38
Improved empowerment and self-confidence	4.57	0.53
Improved care relationship	4.29	1.11
Recommendation to introduce into clinical practice	4.57	0.53

**Table 3 jcm-11-00406-t003:** Evaluation of the digital narrative diary by healthcare professionals.

Healthcare Professionals’ Intermediate Assessment (at 6 Months)	Likert Score (*N* = 5) Mean–SD	
**Feasibility**		
Diary friendliness	4.60	0.55
Diary immediacy and comprehensibility	4.60	0.55
Time management	3.80	0.45
Optimized clinical examination (lenght)	3.20	0.84
Optimized clinical examination (quality)	4.40	0.55
**Utility**		
Improved communication	4.80	0.45
Improved care relationship	4.40	0.55
Deeper knowledge of the patient	4.60	0.55
Improved therapeutic alliance	4.80	0.45
Focus on care history	4.00	0.00
Improved team relationship	4.20	0.84

## Data Availability

The study did not report conclusive data; therefore, there are no details regarding where data supporting the reported results can be found.

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
