# Peer review of "Narrative Medicine: A Digital Diary in the Management of Bone and Soft Tissue Sarcoma Patients. Preliminary Results of a Multidisciplinary Pilot Study"

_jcm, 2022, doi:10.3390/jcm11020406_

Round 1

Reviewer 1 Report

Dear authors,

We read with great interest your article entitled entitled  " Narrative Medicine: a digital diary in the management of bone 2 and soft tissue sarcoma patients. Preliminary results of a 3 multidisciplinary pilot study."

You focus on a pilot study aiming to evaluate the utility of digital diary integrated to the care pathway of patients with sarcomas.

However, there are a range of aspects which have to be addressed. In detail these are as follows:

  • There are no figures or deeper analyses included.
  • It is a very small study - 7 out of 20 enrolled patients completed
  • Results section is 16 lines long
  • Personal statements are nice to have – but no reliable data
  • For sure it is imported to focus more on this topic – but the manuscript will not serve for that or attract attention for that topic
  • The preliminary results and pilot study should be further developed and published with reliable data.

Author Response

Dear Editors,

The authors carefully considered the comments of the reviewers and provided specific replies.

The authors first wish to thank them for considering it important to focus more on narrative-based medicine and for providing expert comments; their comments are valid for the framing of the topic in the scientific panorama as well as for the improvement of the manuscript. The Authors discussed all the reviewers’ comments point by point and revised the manuscript taking into account them. The revisions to the manuscript have been marked using the Word “Track Changes” function.

Two further Authors involved in the data analyses were added; one of them, a native English speaker, checked the language.

Best regards

--------------------------------

The Authors thank this reviewer for his/her expert comments and also for considering the topic worthy of attention. The Authors agree with this reviewer’s opinion that reliable data are expected when considering the title: “preliminary results”.  The authors are aware that, as a qualitative study based on the subjective evaluation of a small group, a preliminary analysis may not be exhaustive and complete, particularly with regard to the point of view of the carers involved; the study design in fact included in an intermediate (at 6 months) and a final health care professionals’ assessment of the tool. Reliable data are still available: the final assessment of the seven patients who completed the study and the intermediate assessment for all the five HCP involved. In conclusion, the authors have accepted the suggestions of this reviewer: a revised version of the manuscript was produced and the limits of the preliminary analysis declared.

Reviewer 2 Report

Mental health and quality of life in bone and soft tissue sarcoma patients is a fundamental question. Clinical Oncologists, Orthopedic Surgeons, and Surgeons must improve the approach to patients' mental health to improve the quality of life. The use of a driven digital diary is a promising idea to rate and work in favor of the patient. However, the authors' type of narrative to present the methods and results needs a more accurate description. The article is a preliminary report of the technique and its potential for improving sarcoma patient care. Still, it is crucial to present a link to any objective quality of life score or another rate method indicating this technique's advantages. The type of narrative is accurate in terms of psychiatry view. However, clinical oncologists and surgeons need a well-defined form to understand the results. A precise cohort description with diagnosis, staging, procedures, and other clinical data is also necessary for clinicians and surgeons to recognize the characteristics of the persons included in the study. That is important, for example, because soft tissue sarcoma surgical treatment is usually less complex than bone sarcoma, which is remarkably critical to the patient's mental health. 
The implementation of the digital dairy occurred in a small size cohort.  The cohort size can be enough to validate a tool but not enough to define the efficacy and the improvement of the quality of life and quality of care. 

Author Response

The Authors thank this reviewer for considering the digital tool a promising idea to work in favor of patients as well as for considering narration a helpful tool, also from a psychiatric point of view. The Authors appreciated this reviewer’s suggestions to improve the quality of the manuscript.

The Authors agree that the digital narrative diary could be suitable to be integrated for the assessment of mental health and for the improvement of quality of life, when applied in psychiatric/psychological clinical practice; even though this could represents a stimulating future evolution of the project, it was not the aim of our pilot study. In fact, taking into account that narrative medicine is a transdisciplinary methodological clinical approach that “reinforces clinical practice with the narrative competencies to recognize, absorb, interpret, and be moved by the stories of illness” (Rita Charon.“Narrative Medicine: Honouring the Story of Illness”2006), the clinicians involved in this study used the diary to improve their routine clinical oncological practice by deepening the knowledge of their patients and strengthening the therapeutic alliance with them.

The Authors, according to this reviewer’s suggestion, improved the manuscript in the methods and results section; reliable data and a cohort description are provide.

Reviewer 3 Report

In their manuscript, Cercato, et al. reported the employment of a digital diary to manage bone and soft tissue sarcoma patients. Given the background that no standard method of narrative medicine is available for the oncological patients in Italy, the authors carried out a pilot study using the “digital narrative diary” to keep track of sarcoma patients. With the application of this narrative medicine tool, they aim to evaluate the benefit of personalized care for sarcoma patients. 20 patients were enrolled in this study and 7 of them completed the final feasibility and utility assessment questionnaire. This is a pilot study and therefore an in-depth analysis is not available. This manuscript further discussed the role of training and education, complexity and multidisciplinarity, and governance in the application of Narrative medicine. To improve the quality of the paper, the authors should improve the following points.

  1. In the text, the authors used a few Italian words. To help international scholars understand better, please translate these Italians words to English when necessary.
  2. In the preliminary results section, the authors described the patients’ information, the highest score in the questionnaire and patients’ comments. However, it lacks an organized and detailed information of these patients and the actual result of the questionnaire. Therefore, it is hard to assess how effective and useful narrative medicine is for sarcoma patients. The authors should at least provide two tables to include such information. First, the overall information of the patients, including their disease type, age, and gender. Second, the authors should list each session of the questionnaire and the scoring from each patient accordingly.

Author Response

The Authors completely agree with all this reviewer’s comments and revised the manuscript according to them. In particular, the methods and results sections have been improved and 3 tables reporting patients’ general characteristics as well questionnaire items and scoring were added.

Round 2

Reviewer 1 Report

Dear Editor, Dear authors,

Thank you very much giving us the opportunity to review the revised version of your manuscript entitled " Narrative Medicine: a digital diary in the management of bone 2 and soft tissue sarcoma patients. Preliminary results of a 3 multidisciplinary pilot study."

Some points are overworked – but we only can repeat: There are no figures or deeper analyses included. From our point of view, there are a lot of points which need to be addressed and included. The preliminary results and pilot study should be further developed and published with reliable data.

Author Response

Dear Reviewer,

This article focuses on the use of a digital diary for the application of narrative medicine in oncological clinical practice, a topic for which small or no data are available. It concerns the preliminary results of a study that used a mixed qualitative and quantitative methodology and that is mainly based on qualitative research criteria. The conclusions of the study are based on the subjective assessments from both the patients’ and the healthcare professionals’ (HCP) perspectives. Tables with results were included. Two factors limited a deeper interim analysis. First, the patients sample size was calculated to reach data saturation (as when no new themes emerge from the data; Morse, 1995); too few patients concluded the study (seven out of 20 enrolled) to generate conclusive hypothesis on the use of the digital diary. Second, according to the study design, HCP assessment was required twice, at 6 months and at the end of the study; the assumption is that the observation of patients during the whole study period is required for HCP’ complete evaluation of the tool.

In conclusion, the Authors opinion is that no further valid data could be available at this point of the study; the Authors’ aim is to present a deeper analysis at the end of the data collection.

Reviewer 2 Report

Considering that it's a pilot study and the improvement of the information, the article is suitable for publication. But narrative style will bring low attraction from surgeons and clinical oncologists.

Author Response

The Authors thank this reviewer for his/her considerations. The Authors are aware that the structure and the narrative style, particularly for the discussion section, is unusual for surgeons and clinical oncologists; however, their opinion is that time is ripe for clinicians to recuperate the humanistic aspects inherent in the art of medicine in order to “personalize” the cure, idea which lies behind the concept of target therapy. Some considerations are probably useful to enlarge the viewpoint, overcome resistances and promote the introduction of patients’ narrations and qualitative research methodology into clinical practice. This is worth trying.

Reviewer 3 Report

After revision, the quality of the manuscript has been greatly improved. The authors have addressed the reviewer's comments well and included more details and clarifications. The paper is suitable for publication if they can carefully check the edited content and correct a few typo. Please use the right symbols when describing the number. For example, in the tables, please use "." instead of "," to describe the decimal. 

Author Response

The Authors thank this reviewer for his/her considerations. The manuscript was checked. Typos and mistakes in the text and tables were corrected and highlighted in yellow in the revised version.